# COVID-19 vaccination in Nigeria: A rapid review of vaccine acceptance rate and the associated factors

**Oluwatosin Olu-Abiodun[1]◉, Olumide Abiodun◉[2]◉*, Ngozi Okafor[1]**

**1** Department of Community Health Nursing, School of nursing Sciences, Babcock University, Ilishan, Ogun State, Nigeria, **2** Department of Community Medicine, School of Clinical Sciences, Babcock University, Ilishan, Ogun State, Nigeria

◉ These authors contributed equally to this work.
* abioduno@babcock.edu.ng

**Data Availability Statement:** This paper is a review of publicly available data. As such, the data underlying the results presented in the study are freely available online. The data used are third party data retrieved from peer-reviewed articles in journals. The exact sources of the data (articles)

## Abstract

Vaccine acceptance among a large population of people can determine the successful control of the COVID-19 pandemic. We aimed to assess the COVID-19 vaccine acceptance rate and to identify the predicting factors to the non-acceptance of the vaccine in Nigeria up to date. In line with this, PubMed, Web of Science, Cochrane Library, and Embase databases were searched for relevant articles between January 2020 and November 2021 in this rapid review. Ten articles with 9,287 individuals met the inclusion criteria and formed the basis for the final COVID-19 acceptance estimates. A total of ten peer-reviewed articles were reviewed. The vaccine acceptance rate ranged from 20.0% to 58.2% among adults across the six geopolitical zones of the country. Non-acceptance of the vaccine was found to be a result of propaganda, adverse effect concerns, and conspiracy theories. National, community, and individual-level interventions need to be developed to improve the COVID-19 vaccine acceptance rate in the country. Greater efforts could be put in place to address the issues of concern leading to the unwillingness of the people to receive the COVID-19 vaccine. Also, as the pandemic is unfolding, emerging evidence needs to be synthesized and updated.

## Introduction

The severe acute respiratory syndrome (SARS) first appeared in November 2002 in the Guang-dong province of southern China [1]. It subsequently spread and resulted in an epidemic of SARS that affected 26 countries with more than 8000 cases in the year 2003 [1]. However, from 2004, no known cases of SARS were reported anywhere in the world [2], until December 2019, when a new strain of severe acute respiratory syndrome coronavirus 2 (SARS-CoV-2) was first reported in Wuhan province of China, the epicenter. The episode of SARS-CoV-2 (now referred to as COVID-19) spread rapidly across the globe and has since been declared by the WHO as a global pandemic. It was first reported in Lagos, Nigeria on February 27, 2020 [3].

underlying our work are listed in the reference section and can be accessed by others at the relevant journal websites. The authors confirm they did not have any special access privilege that others would not have.

**Funding:** 1. Financial disclosure: "Nil" a. The investigators did not receive any institutional or external funding for the study. b. No funder had any role in the study design, data collection and analysis, decision to publish, or preparation of the manuscript. c. The authors did not receive salary from any funding agency for this study. d. The authors received no specific funding for this work.

**Competing interests:** The authors declare no conflict of interest.

As of November 2021, over two hundred and sixty-two million confirmed cases of COVID-19 had been reported globally, with over five million associated deaths and this has led to huge psychological, sociological, and economic turmoil around the globe [4]. In Nigeria, over two hundred and fourteen thousand cases have been reported with more than two thousand deaths [5]. Unfortunately, many Nigerians do not believe in the existence of the virus [6]. At the peak of the pandemic in 2020, far-reaching measures such as extensive testing, nationwide lockdown, social distancing, use of face masks, and isolation of infected persons were put in place to curb the further spread of COVID-19. However, these measures were not enough to contain the spread of the virus and it was not easy to enforce among the general populace.

The coronavirus disease pandemic of 2019 (COVID-19) is humanity's greatest challenge in recent times. Normalcy before the pandemic is unlikely to return until a safe and effective vaccination is successfully implemented. Vaccines strengthen the immune system by using the body's inherent defense mechanisms to boost resistance to specific disease agents [7]. Vaccines generate memory cells, which teach the body's immune infrastructure to rapidly-produce antibodies in the same way that it does when natural infection occurs. Not too long after the emergence of the pandemic, lots of effective and safe vaccines against COVID-19 were rolled out globally. As of the 18th of March 2021, at least thirteen COVID-19 vaccines had received approval for different levels of use, while another twenty-seven were undergoing large-scale, Phase III, final randomized controlled trials [4,8]. Yet, many more are still emerging.

Since there is no specific treatment for COVID-19, vaccination is still one of the most effective means of preventing the disease [6,9]. Table 1 highlights the characteristics of the available COVID-19 vaccines [10]. Despite the progress made by the development of safe and effective vaccines, there are still cogent issues that need to be addressed regarding COVID-19 vaccines.

**Table 1. Common COIVD-19 vaccines [10].**

| Name | Platform | Required doses | Interval btw doses | Efficacy against original strain | Efficacy against variant strains | Prevention of hospital admission | Protection from severe infections | Protection from mild infections |
|---|---|---|---|---|---|---|---|---|
| Pfizer | mRNA in lipid nanoparticles | two | 21 days | 95% | The United Kingdom, South Africa, Latin America | 100% | 100% | 94.1% |
| Moderna | mRNA in lipid nanoparticles | two | 28 days | 95% | The United Kingdom, South Africa, Latin America | 100% | 100% | 95% |
| AstraZeneca/ Oxford | Adenovirus Based | two | Four to 12 weeks | 70% | The United Kingdom, South Africa, Latin America (low) | 100% | 100% | 90% |
| Johnson & Johnson | Non replicating human adenovirus-based/DN | one | NA | 66–77 | The United Kingdom, South Africa, Latin America (low) | 85% | 85% | USA: 72.0% Latin America: 66% South Africa: 57% |
| Sputnik V | Non replicating chimp adenovirus based/DNA | two | 21 Days | 90% | No data | 100% | 100% | USSR: 91.4% |
| Sinovac | Inactivated SARS-CoV-2 | two | 28 days | 50–90% | 50% against Latin America Strain | 100% | - | - |
| Novavax | Protein-based/subunit (RBDMatrix M adjuvant) | two | 16 days | 89% | The United Kingdom, South Africa | 100% | - | UK: 89.3% South Africa: 60% |

Apart from misinformation, disinformation, and anti-vaccine sentiments, there are numerous, carefully designed conspiracy theories around the COVID-19 virus. Besides, vaccine development is constantly challenged by political considerations and religion. This in turn fuels the seemingly retractable non-acceptance of the vaccine [11]. Across the world, it has been reported that people who were fully vaccinated died of COVID-19 associated symptoms, which has also deepened the public uncertainty about the safety and effectiveness of the vaccines [12].

To end the COVID-19 pandemic, an unprecedented call for action at the global, national, and sub-national levels is required. Coupled with restrictive measures like physical distancing and the promotion of healthy behaviors like the wearing of facemasks in public, there is not so much of a choice about the need to obtain the COVID-19 vaccine [13,14]. For the goal of global eradication to be achieved, 70% of all humans must receive the COVID-19 vaccine [15]. In this regard, Nigeria aimed to vaccine 40% of its over 200 million people by the end of the year 2021 and hopes to achieve the 70% vaccination threshold for eliminating COVID-19 before the end of the year 2022 [5].

## COVID-19 vaccination rate

As of 28 November 2021, 7.81 billion vaccine doses had been given globally [4]. In Nigeria, as of 19 November 2021, about six million people had had the initial dose of the COVID-19 vaccine, while only 3,369,628 people had taken the second dose, bringing the unvaccinated population to more than two hundred million, representing 97.15% of the entire population [16].

Despite the urgent and compelling need for the COVID-19 vaccination, considerable COVID-19m vaccine apathy and profound hesitancy are still ingrained in communities. The uptake of the vaccine, therefore, remains low globally and in Africa, especially in Nigeria. At the current rate, Nigeria will likely fall short of its COVD-19 vaccination aspirations. It is unlikely that Nigeria will vaccinate more than 15% of its target. This is particularly worrisome given the prevailing hesitancy among the health workforce and the populations who are most at risk of severe COVID-19 infections (the elderly and people with co-morbidities), who appear to be reluctant to take the jab [6]. If the vaccination target is not met, the epidemic will persist, and continue to cause unnecessary loss of lives, which in turn has implications for the poor morbidity and mortality indices that already exist.

## COVID-19 vaccine acceptance rate

COVID-19 acceptance rates worldwide and in Africa have been surveyed and reported in previous studies. The proportion of willingness to accept COVID-19 vaccination drawn from pooled prevalence acceptance rates across the world has been rated below 60% [17] and 48.93% across Africa [18]. However, some of the problems were not explored. By identifying the factors responsible for unwillingness to accept the vaccine, we can help to inform the development of evidence-based guidelines to effectively improve the COVID-19 uptake. Hence, the current review sought to estimate the COVID-19 acceptance rate and identify the factors responsible for unwillingness to accept the vaccine in Nigeria.

## Materials and methods

This is a rapid review of studies conducted in December 2021. The authors extracted relevant texts from published articles indexed in the African Journals Online (AJOL), Cochrane Review, EMBASE, Google Scholar, HINARI, and PubMed were used to retrieve related articles. During this review, the search was done by using keywords such as; "unwillingness," "acceptance,"," COVID-19 vaccination," "COVID-19," "SARS-CoV-2," "vaccine," and

"Nigeria." The study used the Boolean operators, "AND" and "OR" to incorporate these keywords. Only studies published in the English language between January 2020 and November 2021, and meeting the eligibility criteria were included in this rapid review.

The criteria for eligibility were: (1) peer-reviewed published articles that were indexed in the indicated databases; (2) cross-sectional studies conducted within the general population, or specific population groups like the healthcare workers, students, or family members; (3) studies whose main objective was to assess COVID-19 vaccine acceptance; and (4) manuscript which were published in the English language.

The review, however, excluded; (1) unpublished articles, including preprints; (2) manuscripts whose main objective did not include the assessment of COVID-19 vaccine acceptance; and (3) manuscripts that were published in other languages apart from English. Fig 1 shows the study flow chart.

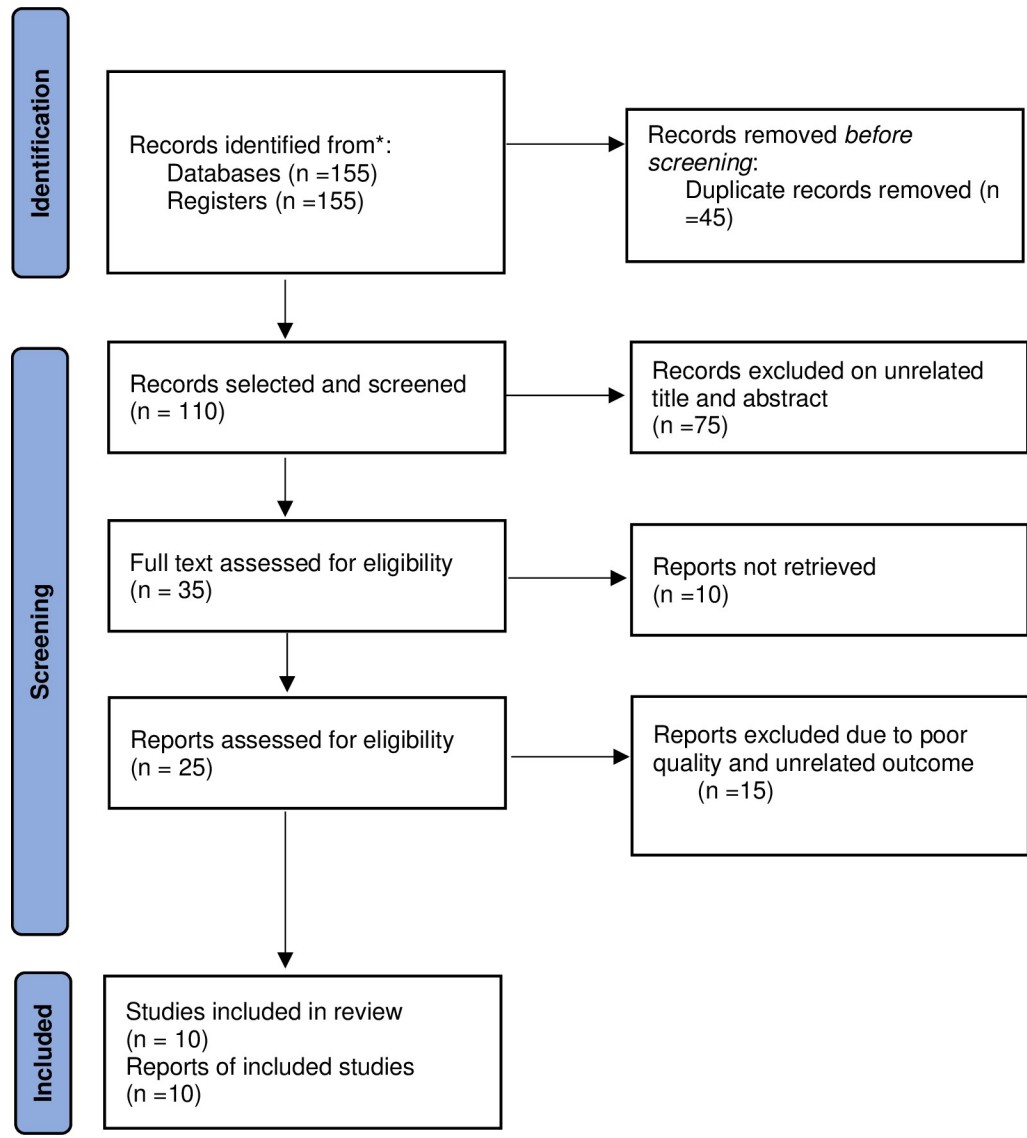

**Fig 1. PRISMA flowchart for COVID-19 vaccine acceptance in Nigeria study search strategy.**

### Ethics statement

This article reports the review of published publicly available manuscripts and did not require the collection of original data or interaction with human participants in any form, therefore, IRB was not obtained. Since the study did not involve participants, consent was neither required nor feasible.

### Data extraction

First, the titles and abstracts were screened, after which data were extracted. The extracted data items were the names of the authors, dates of data collection, the location (particularly, the State) in which the studies were conducted, and the population that was surveyed (for example, the general population, family members, students, etc.). The other items were the sample size, COVID-19 vaccination rates, and the factors that were associated with the COVID-19 vaccine non-acceptance.

## Results

A total of 155 articles were found. The study excluded a total of 45 articles because of duplication, while 75 others had unrelated titles and abstracts, and so were not included in the review. Another 10 were excluded due to the inability to access the full text. The reviewers assessed 25 manuscripts for eligibility, out of which 15 were excluded due to poor quality and unrelated outcome variables. Finally, a total of 10 articles were deemed eligible after meeting the eligibility criteria and, therefore, were included in this review.

### Characteristics of the papers included in this review

The included studies comprised surveys on COVID-19 Vaccine acceptance from across Nigeria with over 11,500 participants, up till March 2021. The survey was mostly done in the southeast of the country. The studies were cross-sectional studies with mainly online recruitment of respondents. The largest sample size was 1740 from a study carried out in Ondo, Edo, and Delta States, while the smallest sample size was 339 from a study carried out in Enugu, Ebonyi, and Anambra States in the South East region. Most of the recruited participants in the studies are adults across the nation. Dates of survey distribution ranged from May 2020 to March 2021.

### COVID-19 vaccine acceptance rate

The results of the COVID-19 vaccine acceptance rates in different studies included in this review are shown in Table 2. The vaccine acceptance rate ranged from 20.0% to 58.2%. Based on the reviewed studies, the highest rates of vaccine acceptance were 58.2% in a study across the six geopolitical zones of Nigeria [19], 55.5% in Ondo, Edo, and Delta [20], 51.1% in Kano [21], and 50.2% across the six geopolitical zones of Nigeria [22]. On the contrary, the lowest acceptance rate was 20.0% across the six geopolitical zones [23], 24.6% from Bayelsa State [24], and 32.52% across the six geopolitical zones of Nigeria [25]. This was followed by 34.7% in Anambra [26], 45.6% in Abia [27], and 47.1% in Abuja [28].

Among health workers, the vaccine acceptance rate was between 32.5 and 55.5%. Out of the three surveys conducted on health workers, two showed COVID-19 vaccine acceptance rates below 50% with the highest of 55.5% in Ondo, Edo, and Delta [20], The lowest COVID-19 vaccine acceptance rate (32.5%) was seen among health workers surveyed in all the six geopolitical zones of the country [25].

Among the adult population, the acceptance rate was between 20.0% and 58.2%. For the five studies conducted among adults, the vaccine acceptance rate was a little above 50% except

**Table 2. Attributes of the eligible studies for the rapid review of COVID-19 vaccine acceptance in Nigeria.**

| s/n | Study | State | Date of survey | Number of participants | Target population | Acceptance rate | Factors |
|---|---|---|---|---|---|---|---|
| 1 | Adejumo OA et al., [20] | Ondo Edo Delta | Oct 2020 | 1740 | Health workers | 55.5% | Vaccines might not be safe, |
| 2 | Uzochukwu IC et al., [26] | Anambra | Jan to Feb 2021 | 349 | University students and staffs | 34.7% | Disbelief, poor knowledge, and understanding of the technology platforms used to design and develop the vaccine, Deficient data about vaccine adverse effect, Religious inclination |
| 3 | Enitan S et al., [23] | Across six geopolitical Zones | May 2020 | 465 | Adults | 20.0% | Disbeliefs, conspiracy theories, and fear of the unknown |
| 4 | Adigwe OP et al., [28] | Abuja | Jan 2021 | 1767 | Adults | 47.1% | Side effects, vaccine safety, and risk concern |
| 5 | Olomofe CO et al., [19] | Across five geopolitical Zones | June to July 2020 | 776 | Adults | 58.2% | Fear of the unknown, conspiracy theories |
| 6 | Tobin EA., et al., [22] | Across 36 States | July to August 2020 | 1228 | Nigerian adults | 50.2% | Misinformation, conspiracy theories, lack of trust in the government, Religious inclination |
| 7 | Amuzie CI et al., [27] | Abia | Mar 2021 | 422 | Health workers | 45.6% | Lack of trust, misinformation, conspiracy theories |
| 8 | Allagoa DO et al., [24] | Bayelsa | Jan to Feb 2021 | 1000 | Patients | 24.6% | Disbelief, conspiracy theories, safety issues, and religious sentiments |
| 9 | Robinson ED et al., [25] | Across six geopolitical zones | Dec 2020 to Jan 2021 | 1094 | Health workers | 32.5% | Effectiveness, fear of the known, and safety concerns. |
| 10 | Iliyasu Z et al., [21] | Kano | Mar 2021 | 446 | Adults | 51.1% | Vaccine safety and rumors |

for the study that had a 20.0% acceptance rate [23]. Among university staff and students, a prevalence rate of 34.7% was reported [26], also, a 24.6% acceptance rate was reported among patients that presented for management at a tertiary health care facility [24].

## Variations in COVID-19 vaccine acceptance rates by population groups over time

Among populations with multiple studies, changes were observed in the COVID-19 acceptance rates over time. Among healthcare workers, the acceptance rate was 55.5% in October 2020, 32.5% in January 2021, and 45.6% in March 2021. Among adults, the acceptance rate was 20.0% in May 2020, 58.2% in July 2020, 50.2% in August 2020, 51.1% in February 2021, and 45.6% in March 2021.

## Factors responsible for non-acceptance of COVID-19 vaccines

The studies explored factors responsible for the refusal of the respondents to accept COVID-19 vaccination. The most stated reasons in the studies were conspiracy theories [19,22–24,27], followed by disbelief [23,24,26], and queried vaccine safety [20,21,24,25,28]. The other reasons are vaccine side effects [21,26,28] and the fear of the unknown [19,23,25].

## Discussion

COVID-19 continues to be a public health concern worldwide and to date. Different strategies have been put in place to combat the pandemic even as new strains of the virus keep emerging.

One of the strategies is the development of the COVID-19 vaccine. Many effective vaccines have been developed and gone through clinical trials why so many are still under trial, yet to be approved. As of today, the COVID-19 vaccine is still the only effective means to control this pandemic and must be accepted by quite a number of the population to be able to combat the pandemic. Therefore, this review was intended to evaluate the acceptance rate of the COVID-19 vaccine among the general population in Nigeria.

This rapid review was done using a comprehensive search to include studies done in Nigeria related to the acceptance rates of the COVID-19 vaccine. It was done using the PRISMA checklist. This is the first review done in Nigeria to assess the acceptance of the COVID-19 vaccine. The findings have critical implications for the government, scientists, policymakers, and program managers. They are also relevant for the communities, and health care providers. From the current review, the estimated pooled prevalence for COVID-19 vaccination acceptance rate among Nigerians ranges between 20.0%- 58.2%. This finding is consistent with surveys that were conducted in most other African countries like Ethiopia (31.4%), Ghana (39.3%), DR Congo (55.9%), and Uganda (53.6%). However, there were substantial differences when compared with studies conducted in Egypt (13.5%), and South Africa (63.3%) [18]. This variation could have resulted from underlying contextual differences, including divergence in the demographic and social features of the different survey populations.

For the COVID-19 vaccine to be termed effective in Nigeria, considerable levels of acceptance are required. Vaccine acceptance has the potential to improve the uptake among the general population and subsequently lead to the eventual development of herd immunity. It has been suggested that about 70% of a population must have immunity either through a vaccine or previous infection for herd immunity to be achieved [29].

This review demonstrated significant variations in the COVID-19 vaccine acceptance rates across different population subgroups. However, certain time-trend patterns can be observed based on when the population groups are compared. Among healthcare workers, the acceptance rate was 55.5% in October 2020, 32.5% in January 2021, and 45.6% in March 2021. Among adults, the acceptance rate was 20.0% in May 2020, 58.2% in July 2020, 50.2% in August 2020, 51.1% in February 2021, and 45.6% in March 2021. It seems that the COVID-19 vaccine acceptance rates first decline, and then began to pick up over time. It is, however, clear that population differences exist in addition to time trends. This picture may be regulated by the study participants' levels of awareness and knowledge of the COVID-19 vaccine at the time of the study.

The factors that were reported for non-acceptance of the COVID-19 vaccination were disbelief, lack of trust in the government, conspiracy theories, vaccine side effects COVID-19 fear of the unknown. These reasons are not surprising as they have been cited in various studies outside Nigeria as reasons for the non-acceptance of vaccines [29,30].

The result of this review demonstrated that the most reported factor in the studies were conspiracy theories that developed from misinformation, fake news, and political sagas radiating the internet in the process of the development of vaccines. In particular, the internet increased the audience for the anti-vaccine movement [31] and has been able to influence a large population against COVID-19 vaccination. General disbelief and lack of trust in the government concerning COVID-19 vaccines were also factors that were reported [32]. Trust in authorities is also associated with vaccination willingness [32]. This might be as a result of the fast production of the vaccines and the process of transaction and procurement of the vaccines from the developed countries that brought about the phrase that "our government accepted them to use us Africans as guinea pigs to test their vaccines [33]. About one million vaccines got expired due to a lack of uptake [34].

This study is the first to review the vaccine acceptance rate and associated factors for the non-acceptance of the COVID-19 vaccine in Nigeria. However, it is a rapid review and is

limited by the relatively few publications on both COVID-19 vaccine acceptance rate and predicting factors that may impact generalizability. There is, therefore a need for a continuous update as new evidence emerges. Even then, the study presents the current state and should help inform research and intervention needs for COVID-19 vaccine acceptance in Nigeria.

## Conclusions and recommendations

Vaccines may be the last hope, for now, to eradicate this deadly virus from the face of the earth. Nevertheless, the majority of the population are nay Sayers, vaccine doubters and so many conspiracy theories circulating that have led to non-acceptance of the vaccine. Hence, to improve vaccine acceptance in Nigeria, context-specific research that is aimed at identifying factors associated with vaccine hesitancy across the prevailing cultural, tribal, and religious tendencies is urgently required [35].

The factors that drive COVID-19 vaccine acceptance, are likely to vary across the different parts of the country. This is the case in the Nigerian context. It is, therefore, unlikely that one single strategy will be used to improve vaccine acceptance in all the populations or regions of the country and the world [35–37]. The capacity of healthcare providers to counter anti-vaccine and anti-science arguments at all levels must be enhanced. This skill is essential because vaccine distrust is established at critical levels in the communities, and often aggravated by highly influential religious leaders. The capacity for effective communication should also be improved to bridge the gap between health workers and the general populace. Targeted interventions for the key populations, including the unvaccinated, under-vaccinated, and hard-to-reach communities, are highly desirable. There should be a deliberate emphasis on community-based approaches to increase awareness and knowledge of the COVID-19 vaccine [35]. It is essential to devise innovative theory-based interventions to engage critical stakeholders like community chiefs, religious leaders, and others to enhance the community-based COVID-19 vaccination drive. Also, to combat the widespread rumors and misinformation that has led to the non-acceptance of the COVID-19 vaccine, effective health messaging campaign must be put in place to encourage the acceptance of the vaccine. Policymakers, healthcare workers, and other stakeholders need to do more about information dissemination and health education and promotion, especially on the misconceptions about the COVID-19 vaccine.

## Supporting information

**S1 Checklist.**
(DOCX)

## Author Contributions

**Conceptualization:** Oluwatosin Olu-Abiodun.

**Data curation:** Oluwatosin Olu-Abiodun, Olumide Abiodun.

**Investigation:** Oluwatosin Olu-Abiodun.

**Methodology:** Oluwatosin Olu-Abiodun.

**Project administration:** Olumide Abiodun.

**Resources:** Ngozi Okafor.

**Supervision:** Olumide Abiodun.

**Validation:** Olumide Abiodun.

**Visualization:** Ngozi Okafor.

**Writing – original draft:** Oluwatosin Olu-Abiodun.

**Writing – review & editing:** Oluwatosin Olu-Abiodun, Olumide Abiodun, Ngozi Okafor.

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
