## [Decision Letter · Decision Letter 0]

15 Feb 2022

PONE-D-21-39940COVID-19 vaccination in Nigeria: a rapid review of vaccine acceptance rate and associated factorsPLOS ONE

Dear Dr. Olumide,

Thank you for submitting your manuscript to PLOS ONE. After careful consideration, we feel that it has merit but does not fully meet PLOS ONE’s publication criteria as it currently stands. Therefore, we invite you to submit a revised version of the manuscript that addresses the points raised during the review process.

We look forward to receiving your revised manuscript.

Kind regards,

Nusirat Elelu

Academic Editor

PLOS ONE

Journal Requirements:

- https://www.njgp.org/article.asp?aulast=Ogundele&epage=4&issn=1118-4647&issue=1&spage=1&volume=18&year=2020

- https://pubmed.ncbi.nlm.nih.gov/34616860/

In your revision ensure you cite all your sources (including your own works), and quote or rephrase any duplicated text outside the methods section. Further consideration is dependent on these concerns being addressed.

 “Nil”

5. Please ensure that you refer to Figure 1 in your text as, if accepted, production will need this reference to link the reader to the figure.

6. We note you have included a table to which you do not refer in the text of your manuscript. Please ensure that you refer to Table 1 in your text; if accepted, production will need this reference to link the reader to the Table.

Additional Editor Comments:

Dear Author,

Your manuscript has been reviewed and was found to provide information on Covid-19 acceptance in Nigeria, however there is need to improve the quality of the manuscript by citing references to back up statements made and finding from the survey. There were also a lot of punctuation, grammatical inconsistencies that needs to be improved upon.

Kindly exhaustively look through comments made by reviewers and make necessary correction before you resubmit for further consideration.

Reviewers' comments:

Reviewer's Responses to Questions

**Comments to the Author**

1. Is the manuscript technically sound, and do the data support the conclusions?

Reviewer #1: Yes

Reviewer #2: Yes

2. Has the statistical analysis been performed appropriately and rigorously? 

Reviewer #1: N/A

Reviewer #2: N/A

3. Have the authors made all data underlying the findings in their manuscript fully available?

Reviewer #1: Yes

Reviewer #2: Yes

4. Is the manuscript presented in an intelligible fashion and written in standard English?

Reviewer #1: Yes

Reviewer #2: Yes

5. Review Comments to the Author

Reviewer #1: COVID-19 vaccination in Nigeria: a rapid review of vaccine acceptance rate and associated factors

This study conducted a rapid review of the acceptance rate of the COVID-19 vaccine as well as identify factors affecting the acceptance rate among the Nigerian populace using reports from 10 published article. Vaccine acceptance rate recorded was not more than 59%. Vaccine hesitancy was due to disbelief, propaganda, adverse effect concerns, and conspiracy theories.

This manuscript is no number-lined and thus made the review to be difficult.

Abstract

There is the use of clutters affecting the conciseness of the abstract presentation.

The authors should include the duration of the web search for relevant publications.

Suggestions for future research were not mentioned.

There is an inconsistency in the way COVID-19 is written in the abstract and other parts of the manuscript. Authors should check and present the appropriate presentation.

Introduction

First line: start the sentence with the article “The”.

Lines 2 and 3 of the introduction: the sentence is not presented clearly. Authors should re-cast to make better meaning.

The third sentence of the first paragraph of the introduction is too long and this blurs the meaning of the information the authors are trying to convey. The authors should break-up the sentence into smaller and meaningful sentences.

Authors should include citation(s) to this statement made in the second paragraph of the introduction “Unfortunately, many Nigerians do not believe in the existence of the virus”.

Please include “in recent time” as part of the first sentence of the third paragraph of the introduction.

Please include citation(s) to this statement found also in the third paragraph of the introduction “Vaccines strengthen the immune system by utilizing the body's defenses to build resistance to specific pathogens”.

Authors should include some citations to paragraph four and five of the introduction. A lot of categorical statements were made requiring citations. This should also be done for the paragraph on COVID-19 vaccination rate.

Materials and methods

Authors should state the specific time duration used for this cross-sectional survey.

Figure 1: The title is not descriptive enough.

Figure 1 was not cited anywhere within the manuscript.

Results and Discussion

Under “characteristics of the papers included in this review”, the authors should pluralize “state” anytime the word is ascribed to more than one states in Nigeria.

Generally, the authors have easily captured the results obtained in entirety. The discussion could be made more robust by using references especially from Africa and other regions of the world with the vaccine acceptance rates and factors affecting these rates well discussed. The COVID-19 landscape is changing rapidly and authors should do more to make this work more robust.

Punctuation errors were noticed in the discussion and other parts of the manuscript.

Limitation to the research were presented and discussed. However, the use of limited articles in this review really affected the validity of the outcomes. This could possible due to the short duration of the study period. Authors should extend its search beyond the current periods to accommodate more published articles in the Nigerian context thus could led to the presentation of a more valid outcome.

Reviewer #2: This is a good and relevant review manuscript on COVID-19 vaccination in Nigeria: a rapid review of vaccine acceptance rate and associated factors.

All comments have been included in the attached manuscript for necessary corrections. References should cited where indicated to support some of the points.

6. PLOS authors have the option to publish the peer review history of their article (what does this mean?). If published, this will include your full peer review and any attached files.

Reviewer #1: No

Reviewer #2: **Yes: **Dr. AbdulAzeez A. Anjorin

---

## [Author Response · Author response to Decision Letter 0]

7 Mar 2022

Dear Editor for PlosOne,

Thank you for the prompt and thorough review of our manuscript. We acknowledge the efforts of the academic editor and reviewers, and appreciate them immensely. We have taken note of all the suggestions and requirements by your team. Our overall attitude is to accept the requests of the reviewers and make the corrections. We are of the opinion that this has greatly improved the quality of our manuscript and, therefore, believe that it will receive a more favourable review on this occasion.

We present the details of the adjustments made to the paper on a point-to-point basis below.

Editorial comments

This has been strictly adhered to.

This is most unfortunate. We have now resolved those issues comprehensively.

3. Thank you for stating the following financial disclosure: Nil

We have included the appropriate statements in the cover letter.

We have included the appropriate statements in the cover letter.

5. Please ensure that you refer to Figure 1 in your text as, if accepted, production will need this reference to link the reader to the figure.

We have referred to this figure in the paragraph just above where it is located.

6. We note you have included a table to which you do not refer in the text of your manuscript. Please ensure that you refer to Table 1 in your text; if accepted, production will need this reference to link the reader to the Table.

We have referred to this table in the paragraph just above where it is located.

Additional Editor Comments:

1. There is need to improve the quality of the manuscript by citing references to back up statements made and finding from the survey. 

Additional references have been included where necessary.

2. There were also a lot of punctuation, grammatical inconsistencies that needs to be improved upon.

A thorough editing to correct for punctuation and grammatical inconsistencies has be done.

Reviewer #1

1. This manuscript is no number-lined and thus made the review to be difficult.

Apologies. We are not aware that the submission requires line numbering. However, we have double-spaced the text and hope that this makes reading easier.

Abstract

2. There is the use of clutters affecting the conciseness of the abstract presentation.

3. The authors should include the duration of the web search for relevant publications.

“databases were searched for relevant articles between January 2020 and November 2021”

4. Suggestions for future research were not mentioned.

“Also, as the pandemic is unfolding, emerging evidence needs to be synthesized and updated.”

5. There is an inconsistency in the way COVID-19 is written in the abstract and other parts of the manuscript. Authors should check and present the appropriate presentation.

We have rectified the inconsistencies. It is written as COVID-19 all through the manuscript now.

Introduction

1. First line: start the sentence with the article “The”.

Done.

2. Lines 2 and 3 of the introduction: the sentence is not presented clearly. Authors should re-cast to make better meaning.

The sentence read better now: “It subsequently spread and resulted in an epidemic of SARS that affected 26 countries with more than 8000 cases in the year 2003”.

3. The third sentence of the first paragraph of the introduction is too long and this blurs the meaning of the information the authors are trying to convey. The authors should break-up the sentence into smaller and meaningful sentences.

The sentences read better now: “However, from 2004, no known cases of SARS were reported anywhere in the world (2), until December 2019, when a new strain of severe acute respiratory syndrome coronavirus 2 (SARS-CoV-2) was first reported in Wuhan province of China, the epicenter. The episode of SARS-CoV-2 (now referred to as COVID-19) spread rapidly across the globe and has since been declared by the WHO as a global pandemic”.

4. Authors should include citation(s) to this statement made in the second paragraph of the introduction “Unfortunately, many Nigerians do not believe in the existence of the virus”.

An appropriate citation has been included: “Adesegun O, Binuyo T, Adeyemi O, Ehioghae O, Rabor D, Amusan O, et al. The COVID-19 Crisis in Sub-Saharan Africa: Knowledge, Attitudes, and Practices of the Nigerian Public. The American Journal of Tropical Medicine and Hygiene. 2020;103(5):1997-2004”.

5. Please include “in recent time” as part of the first sentence of the third paragraph of the introduction.

Done.

6. Please include citation(s) to this statement found also in the third paragraph of the introduction “Vaccines strengthen the immune system by utilizing the body's defenses to build resistance to specific pathogens”.

An appropriate citation has been included: “Clem AS. Fundamentals of vaccine immunology. Journal of global infectious diseases. 2011;3(1):73”.

7. Authors should include some citations to paragraph four and five of the introduction. A lot of categorical statements were made requiring citations. This should also be done for the paragraph on COVID-19 vaccination rate.

Appropriate citations have been included:

“Koirala A, Joo YJ, Khatami A, Chiu C, Britton PN. Vaccines for COVID-19: The current state of play. Paediatric Respiratory Reviews. 2020;35:43-9”.

“Schoch-Spana M, Brunson EK, Long R, Ruth A, Ravi SJ, Trotochaud M, et al. The public’s role in COVID-19 vaccination: Human-centered recommendations to enhance pandemic vaccine awareness, access, and acceptance in the United States. Vaccine. 2021;39(40):6004-12”.

“Nuzhath T, Tasnim S, Sanjwal RK, Trisha NF, Rahman M, Mahmud SF, et al. COVID-19 vaccination hesitancy, misinformation and conspiracy theories on social media: A content analysis of Twitter data”. 2020.

“Vallis M, Bacon S, Corace K, Joyal-Desmarais K, Sheinfeld Gorin S, Paduano S, et al. Ending the Pandemic: How Behavioural Science Can Help Optimize Global COVID-19 Vaccine Uptake. Vaccines. 2022;10(1):7”.

“Guimón J, Narula R. Ending the COVID-19 Pandemic Requires More International Collaboration. Research-Technology Management. 2020;63(5):38-41”.

“Irwin A, Nkengasong J. What it will take to vaccinate the world against COVID-19. Nature. 2021;592(7853):176-8”.

Materials and Methods

1. Authors should state the specific time duration used for this cross-sectional survey

This has been stated clearly: “This is a rapid review of studies conducted in December 2021”

2. Figure 1: The title is not descriptive enough.

The title has been enhanced: “PRISMA flowchart for COVID-19 vaccine acceptance in Nigeria study search strategy”.

3. Figure 1 was not cited anywhere within the manuscript.

Figure 1 has now been cited in the paragraph preceding its location in the manuscript.

Results and discussion

1. Under “characteristics of the papers included in this review”, the authors should pluralize “state” anytime the word is ascribed to more than one states in Nigeria.

The relevant portions have been pluralized.

2. Generally, the authors have easily captured the results obtained in entirety. The discussion could be made more robust by using references especially from Africa and other regions of the world with the vaccine acceptance rates and factors affecting these rates well discussed. The COVID-19 landscape is changing rapidly and authors should do more to make this work more robust.

We have included more references from African and other studies. The discussion has been generally improved to make it more robust and capture the dynamic trends.

Wake AD. The acceptance rate toward COVID-19 vaccine in Africa: a systematic review and meta-analysis. Global Pediatric Health. 2021;8:2333794X211048738.

“Gakpo J. We Africans don’t want to be guinea pigs for Covid-19 vaccines, but who should? Accra: GhanaWeb; 2020 [updated 29th January 2022. Opinions]. Available from: https://www.ghanaweb.com/GhanaHomePage/features/We-Africans-don-t-want-to-be-guinea-pigs-for-Covid-19-vaccines-but-who-should-937816”.

“Shepherd A. Vaccines wasted as Africa waits. British Medical Journal Publishing Group; 2022”.

"Ogundele O, Ogundele T, Beloved O. Vaccine hesitancy in Nigeria: Contributing factors -way forward. The Nigerian Journal of General Practice. 2020;18(1):1-4”.

“MacDonald NE, Butler R, Dubé E. Addressing barriers to vaccine acceptance: an overview. Human Vaccines & Immunotherapeutics. 2018;14(1):218-24”.

3. Punctuation errors were noticed in the discussion and other parts of the manuscript.

We have thoroughly edited the manuscript to remove these errors.

4. Limitation to the research were presented and discussed. However, the use of limited articles in this review really affected the validity of the outcomes. This could possible due to the short duration of the study period. Authors should extend its search beyond the current periods to accommodate more published articles in the Nigerian context thus could led to the presentation of a more valid outcome.

This is a potent argument and is noted in our discussion. The pandemic is evolving and emerging data will validate the findings. However, the study aims to present the state of evidence to provide a basis for further investigation, update, and interventions. Interventions are required NOW! We have in our recommendations pointed to need for ongoing updates.

Reviewer #2

1. This is a good and relevant review manuscript on COVID-19 vaccination in Nigeria: a rapid review of vaccine acceptance rate and associated factors.

Thanks for the kind comments.

2. All comments have been included in the attached manuscript for necessary corrections. References should be cited where indicated to support some of the points.

We have made all the adjustments requested within the manuscript and they have enriched our paper. We also included the relevant citations in the appropriate sections as requested. The following fresh citations were included;

“Wake AD. The acceptance rate toward COVID-19 vaccine in Africa: a systematic review and meta-analysis. Global Pediatric Health. 2021;8:2333794X211048738’.

“Gakpo J. We Africans don’t want to be guinea pigs for Covid-19 vaccines, but who should? Accra: GhanaWeb; 2020 [updated 29th January 2022. Opinions]. Available from: https://www.ghanaweb.com/GhanaHomePage/features/We-Africans-don-t-want-to-be-guinea-pigs-for-Covid-19-vaccines-but-who-should-937816”.

“Shepherd A. Vaccines wasted as Africa waits. British Medical Journal Publishing Group; 2022”.

"Ogundele O, Ogundele T, Beloved O. Vaccine hesitancy in Nigeria: Contributing factors -way forward. The Nigerian Journal of General Practice. 2020;18(1):1-4”.

“MacDonald NE, Butler R, Dubé E. Addressing barriers to vaccine acceptance: an overview. Human Vaccines & Immunotherapeutics. 2018;14(1):218-24”.

“Adamu Aa, Essoh T-A, Adeyanju GC, Jalo RI, Saleh Y, Aplogan A, et al. Drivers of hesitancy towards recommended childhood vaccines in African settings: a scoping review of literature from Kenya, Malawi and Ethiopia. Expert Review of Vaccines. 2021;20(5):611-21”.

---

## [Editor Report · Decision Letter 1]

14 Apr 2022

COVID-19 vaccination in Nigeria: a rapid review of vaccine acceptance rate and the associated factors

PONE-D-21-39940R1

Dear Dr. Abiodun_Olumide

We’re pleased to inform you that your manuscript has been judged scientifically suitable for publication and will be formally accepted for publication once it meets all outstanding technical requirements.

Kind regards,

Nusirat Elelu

Academic Editor

PLOS ONE
---

## [Editor Report · Acceptance letter]

3 May 2022

PONE-D-21-39940R1 

COVID-19 vaccination in Nigeria: a rapid review of vaccine acceptance rate and the associated factors 

Dear Dr. Abiodun:

I'm pleased to inform you that your manuscript has been deemed suitable for publication in PLOS ONE. Congratulations! Your manuscript is now with our production department. 

Kind regards, 

on behalf of

Dr. Nusirat Elelu 

Academic Editor

PLOS ONE